# Vegetation Drastically Reduces Wind Erosion: An Implementation of the RWEQ in the Mongolian Gobi Steppe

**Isita Nandana Talukdar [1],\*, Virginia Anne Kowal [2], Binbin Huang [3] and Charlotte Weil [4]**

1   5269 Manderston Drive, San Jose, CA 95138, USA
2   Natural Capital Project, Woods Institute for the Environment, Stanford University, 327 Campus Drive, Bass Biology Building 123, Stanford, CA 94305, USA; ginger.kowal@gmail.com
3   State Key Laboratory of Urban and Regional Ecology, Research Center for Eco-Environmental Sciences, Chinese Academy of Sciences, 18 Shuangqing Road, Haidian District, Beijing 100085, China; bbhuang_st@rcees.ac.cn
4   ENAC School of Architecture, Civil and Environmental Engineering, EPFL, Route Cantonale, CH-1015 Lausanne, Switzerland; charlotte.weil@epfl.ch
\*   Correspondence: isitatalukdar@gmail.com; Tel.: +1-669-400-6142

**Abstract:** Soil loss prevention is an important ecosystem service for protecting human and environmental health. Using spatiotemporal climate and environmental data of the Eastern Gobi Steppe, a region missing from previous studies of Mongolian wind-based soil erosion, we implemented the Revised Wind Erosion Equation (RWEQ) model to estimate soil loss. A replicable pipeline was developed to perform these computations, and made available openly. Soil loss was estimated on a monthly basis to analyze seasonal variations. The results show that the annual total soil loss was $61 \times 10^{10}$ kg over an area of $69.3 \times 10^3$ km$^2$, which is about 90 tonnes per hectare. Increasing fractional vegetation coverage to a uniform 50% coverage (doubling current vegetation coverage in every 1 km$^2$) could reduce soil loss by 60%, highlighting the importance of protecting and increasing vegetation coverage in ecosystem service preservation.

**Keywords:** wind erosion; soil loss; ecosystem services; revised wind erosion equation; Gobi Steppe

## 1. Introduction

Soil loss due to wind erosion is a grave problem for semiarid and arid areas, where the lack of precipitation and scant vegetation coverage cannot prevent surface materials from being blown away by strong and frequent winds [1]. This can lead to desertification, to severe soil coarseness, infertility, and dryness, which affect the ability to grow crops for sustenance [2,3]. Wind erosion also creates sandstorms, and air pollution which affect pulmonary health [4]. A study done in Phoenix, Arizona, USA, found "that the cropland wind erosion was responsible for 55% and 51% of total PM$_{10}$ over Phoenix", with PM$_{10}$ meaning dust particulate matter with diameter 10 μm and smaller including that caused by wind erosion [5]. Another study in Slovakia concluded that soil particles from up to a depth of 5 cm were displaced by wind, with the soil surface becoming "much rougher" [6]. Indeed, the soil particles that enter the air from sandstorms are easily inhalable and can cause significant respiratory ailments due to oxidative stress, as they are embedded into deep lung tissue and the subepithelial environment [7,8]. Such dust particulates can also increase the rates of anemia, liver and kidney disease, and asthma [9]. For all these reasons, soil erosion control is one of the fundamental ecosystem services for human welfare [10].

Wind erosion is caused by a combination of climate factors, such as wind speeds, temperature, precipitation, and soil factors such as surface terrain roughness, and vegetation coverage. The Wind Erosion Equation (WEQ) published by Woodruff and Siddoway [11] models this phenomenon. The Revised Wind Erosion Equation (RWEQ) extends the WEQ by utilizing wind as the primary factor rather than soil erodibility as WEQ does.

Fryrear et al. [12] showed that the RWEQ is a much more accurate model: the correlation between measured erosion and estimated erosion with WEQ was not significant ($r^2 = 0.01$), whereas the correlation with RWEQ was highly significant ($r^2 = 0.927$) [12]. The RWEQ also takes soil surface roughness into account, unlike the WEQ.

We used the RWEQ model to estimate wind erosion in the Gobi Steppe. In East Asia, the Gobi Steppe is a region sensitive to global climate change, due to degraded grasslands and deserted croplands [13]; where dust storms are regarded as one of the most severe environmental problems [14]. Our area of interest specifically is the Eastern Gobi Steppe.

Closer to the Eastern Gobi Steppe, in the Inner Mongolia Province of China, wind erosion had been simulated using the RWEQ by Jiang et al. [15], where they have shown that there is high correlation ($r^2 = 0.88$) between RWEQ estimates of wind erosion and measurements obtained from remote sensing data [15]. They found that 23.2% of this erosion could be rated as severe between 2001 and 2010 [15]. Corroborating these findings, Zhang [16] found that 23.2% of the wind erosion can be rated as severe from 1990 to 2015, also using RWEQ-based models. Other approaches resulted in more variability—e.g., Qi [14] found annual soil erosion rates ranging from 53.12 t/km$^2$ to 479.63 t/km$^2$ over eight sampling sites, using the Cs tracing technique [17], which analyzes changes of 137-Caesium, which is found bound to soil colloids, in soil to measure soil erosion on 8 sampling sites in the Mongolian Plateau; Shi [18] found a categorical map of wind erosion across the Mongolian Plateau, ranking wind erosion from none to severe, using a fuzzy c-means clustering technique which is an automatic classification method that expresses gradual spatial changes of physical geographic phenomena, such as wind erosion in this case, by using a continuously divided membership model [18]. A more recent study utilized the RWEQ in Inner Mongolia to calculate soil loss and categorize it according to standards of the Ministry of Water Resources of China, and analyzed the migration paths of the erosion, finding that: "Tolerable and slight aeolian erosion intensities characterized most of Inner Mongolia's area, being widely distributed across the region, while moderate intensity was mainly clustered in its southwest and southeast" and that "the SL's maximum, minimum, and mean values sharply decreased prior to 2013 but they increased later" [19]. Another recent study utilized the RWEQ to calculate wind erosion in Inner Mongolia, finding that wind erosion intensity in many regions increased from light to mild and moderate, with the area and intensity of soil wind erosion displaying an increasing trend [20]. Though the RWEQ has been validated in these areas surrounding the Eastern Gobi Steppe, no ground truth measurements are available, to date, in the Eastern Gobi Steppe itself.

The Chinese government implemented measures to prevent sandstorms and wind erosion via the Beijing−Tianjin Sandstorm Source Control Project to be implemented from 2013 to 2022 [21]. This project is focused on the Chinese provinces of Beijing, Tianjin, Hebei, Shanxi, Shaanxi, and Inner Mongolia. In this project, resources are invested into afforestation, sand fixation, and degraded shelter forest reconstruction. Afforestation, the planting of new trees, and forest reconstruction, increase vegetation coverage significantly. The benefits of this project are widespread. For example, the income of farmers and herdsmen in the area has been increased, and the more stable ecosystem bolstered by the increase in vegetation coverage will contribute to conservation of animal and plant species.

Though wind erosion has been estimated in the Inner Mongolia Province in China, by Jiang et al., [15] and Zhang et. al., [16], estimates are missing for the Eastern Gobi Steppe. Based on the strong correlation of RWEQ estimates against wind erosion measurements shown by Fryrear et. al. [12] and Jiang et. al. [15], we aim here to use a RWEQ-based model to estimate wind erosion in the Mongolian Gobi Steppe and, most importantly, understand its main drivers. This will allow us to simulate wind erosion under possible future scenarios of climate and vegetation cover. This work also outputs a replicable implementation of the RWEQ-based wind erosion model, globally applicable with data at various resolutions.

*The Eastern Gobi Steppe*

The Eastern Gobi Steppe stretches from the Inner Mongolia Plateau in China into Mongolia at 1000 to 1500 m elevation, northward into Mongolia. Summer heat depends on elevation, ranging from warm to hot, and winters are severely cold. Winter conditions are harsher here than other parts of China at similar altitudes and latitudes, as there are no mountains to shelter the region from northerly winds. Annual total precipitation varies significantly, and most of the precipitation is in summer. Vegetation consists of drought-resistant shrubs and low grasses.

Our study area, an area of $69.3 \times 10^3$ km², where sensitivity analysis was conducted, is located along Mongolia's southern border and inside the Gobi desert. This area is characterized by extremely low rainfall and very little vegetation cover [22]. The mean temperature is 3.7 °C annually, with a maximum of 31.5 °C and a minimum of −23.2 °C. Annual precipitation is 100 to 150 mm, and most rainfall occurs in summer [23].

We used the RWEQ model to estimate wind erosion in one region of the Mongolian Plateau (Figure 1). In East Asia, the Mongolian Plateau is a region sensitive to global climate change, where dust storms are regarded as one of the most severe environmental problems [14]. We conducted a sensitivity analysis of the RWEQ model in a portion of the Eastern Gobi Steppe, outlined in the map with a red border, to analyze which environmental factors have the greatest influence in predicting wind erosion. The results of the analysis are detailed in further sections of this paper.

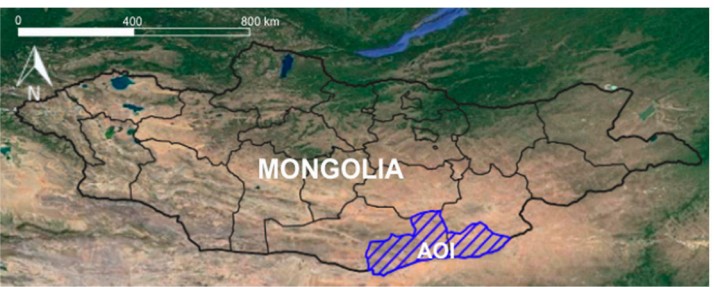

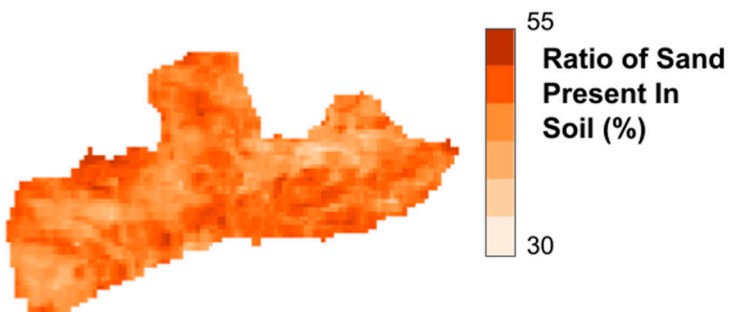

**Figure 1.** Map of our Area Of Interest (AOI) in Mongolia and map of percentage sand content in the study area. Borders overlayed on Google Earth satellite base map.

## 2. Methods

### 2.1. RWEQ Model

The RWEQ model estimates soil loss (*SL*) for a given location. Weather, soil erodibility, soil crust, surface terrain roughness, and vegetation coverage factors are used to calculate the maximum transported capacity by wind (*Qmax*) and the critical field length (*S*) as shown below [24,25]:

$$Qmax = 109.8 * (WF * EF * K\prime * SCF * COG) \tag{1}$$

$$S = 150.71 * (WF * EF * K\prime * SCF * COG)^{-0.3711} \tag{2}$$

$$SL = 100 / \left( S^2 + 0.01 \right) * Qmax * \left( e^{-(50/(S+0.01))} \right)^2 \tag{3}$$

where *Qmax* (in kg/m) is the transport capacity and *S* (in m) is the distance where 63% of the maximum transport capacity is reached, called the critical field length. *SL* (in kg/m$^2$) is the rate of soil loss caused by wind erosion (Z from the original equation is 50 here, and is already factored in). *WF* is the weather factor, *SCF* is the foil crusting Factor, *EF* is the soil erodible fraction, *COG* is the vegetation factor, and *K'* is the surface terrain roughness factor.

**Weather factor (*WF*, in kg m$^{-1}$).** Wind is the cornerstone of RWEQ, with soil moisture and snow cover as important factors influencing wind erosion. In this paper, the weather factor is calculated on a monthly basis. The weather factor represents the effect of climate on wind erosion, and combines wind speed, soil moisture, and snow cover as shown below:

$$WF_{monthly} = Wf * \rho / 9.8 * SW * SD \tag{4}$$

where *WF* is the weather factor (in kg/m), *Wf* is the wind factor (in m$^3$/s$^3$), $\rho$ is air density (in kg/m$^3$), 9.8 is acceleration due to gravity (in m/s$^2$), SW represents soil moisture, and *SD* represents snow cover.

The wind factor is calculated from daily wind speed (*ws*, in m/s) using the equation shown below [25] when the wind speed value is higher than the threshold of 5 m/s. For all wind speed values less than this threshold, the wind factor is equal to 0 [26]. Indeed, the stronger the wind, the more soil matter will be transported:

$$Wf = ws * (ws - 5)^2 \tag{5}$$

The soil moisture factor is calculated from monthly potential evapotranspiration (*ETp*, in mm), monthly total precipitation (*PRCP*, in mm), monthly number of rainy days (*PRCP_DAYS*) as shown below [25]. Increased rain decreases the soil moisture factor, which decreases wind erosion:

$$SW = (ETp - PRCP * PRCP\_DAYS) / ETp \tag{6}$$

Monthly potential evapotranspiration is calculated from monthly total solar radiation (*SOL*, in MJ/m$^2$) and monthly average temperature (*TEMP*, in °C) as shown [27]. Lower solar radiation and lower temperatures lead to lower evapotranspiration, which in turn decreases erosion:

$$ETp = 0.0135 * (SOL/2.54) * (TEMP + 17.8) \tag{7}$$

*SD* is the snow cover factor determined as 1—probability of snow depth > 25.4 mm [25].

**Soil crusting factor (*SCF*).** When rain hits the soil surface, there is a formation of surface crust because of redistributed soil particles. That soil surface can be hard or weak and may affect wind erosion potential [15]. The soil crusting factor is a non-temporal value calculated from clay (*CL*, %) and organic matter (*OM*, %) as shown below [15]. Greater percentage of clay and organic matter makes the soil less susceptible to wind erosion:

$$SCF = 1 / \left( 1 + 0.0066 * CL^2 + 0.021 * OM^2 \right) \tag{8}$$

**Soil erodible fraction (*EF*).** The erodible fraction is the fraction of the surface 25 mm of soil that is lower than 0.84 mm in diameter as determined by a standard compact rotary sieve, and the highest value of *EF* for a year is related to the physical and chemical properties of the soil [15]. The erodible fraction is a non-temporal value calculated from the ratios of sand (*SND*, %), silt (*SILT*, %), clay (*CL*, %), and organic matter (*OM*, %) in the soil as shown in the following equation [25]. Indeed, a larger percentage of sand and silt increase wind erosion as the soil is more easily transported by wind:

$$EF = (29.09 + 0.31 * SND + 0.17 * SILT + 0.33 * SND/CL - 2.59 * OM - 0.95 * 0)/100 \tag{9}$$

**Vegetation factor (*COG*).** The amount of vegetation coverage over the soil can have a significant impact on how well the soil is protected from erosion. An increased percentage of vegetation coverage decreases the soil's ability to be eroded by wind. The monthly vegetation factor can be calculated from the monthly fraction of vegetation coverage (*FVC*, %) as shown in the following equation [25]:

$$COG = e^{(-0.00438*FVC)} \tag{10}$$

The original equation has additional factors as shown in Jarrah [25]. However, due to limited availability of data, only the term related to *FVC* was used. This was based on the work done by Jiang [15] in the same geographical region:

$$COG = e^{(-0.00438*FVC)} + e^{(-0.0344SA^{0.6413})} + e^{(5.614CC^{0.7366})} \tag{11}$$

where *FVC* is the percentage of land covered by crop residue, *SA* is the standing stem area index, and *CC* is the percentage soil surface covered by crop canopy.

**Surface terrain roughness factor (*K'*).** Surface terrain roughness is calculated on a monthly basis from topographical roughness (*Kr*) and the chain random roughness (*Crr*), a measure of surface roughness via the chain method [28], as shown in the following equation [15]:

$$K' = e^{(1.86*Kr-2.41*Kr^{0.934}-0.127Crr)} \tag{12}$$

Topographical roughness is calculated for $3 \times 3$ grid of pixels of the raster data from the difference between the minimum and maximum elevations in the selected area (*H*, in m) and the length of the edge of the area ($L = 3 \times$ resolution of raster) in the following equation [15]. Indeed, the topographic roughness offers protection against erosion:

$$Kr = 0.2 * \left(H^2/L\right) \tag{13}$$

Chain random roughness is calculated from the monthly fraction of vegetation coverage (*FVC*) as shown in the following relationships. It is computed in two steps. First, a random roughness is computed from *FVC* [29]. Then the computed random roughness (*RR*) is used to calculate chain random roughness (*Crr*) [30]. Chain random roughness similarly decreases wind erosion:

$$RR = 0.025 + 2.464FVC^{3.56} \tag{14}$$

$$Crr = 17.46 * (RR)^{0.738} \tag{15}$$

*2.2. Input Data Sources*

The following input data is utilized to calculate soil erosion based on the RWEQ equations and direction shown in Figure 2. The digital elevation model (DEM) data were downloaded from the Shuttle Radar Topography Mission (SRTM) website for the country of Mongolia [31]. Wind speed was sourced from the ERA5-Land Dataset on the Copernicus Climate Data Store [32]. Monthly average temperature and total precipitation came directly from Worldclim current conditions [33]. Solar radiation was calculated according to Penman [34]. Precipitation days information from a public website was used [35]. To justify the use of this website, the data were compared to Climate Hazards Group InfraRed Precipitation with Station data (CHIRPS), and found to be only 1 or 2 days apart. As a varying number of rain days was shown to have almost no impact on wind erosion results based on the sensitivity analyses, the travel website data served as a reasonable source of the number of rain days. Clay, silt, and sand fractions were from ISRIC SoilGrids 250 m [36]. Soil organic matter and monthly snow cover are outputs of the Rangeland Production Model [37], driven by Worldclim historical vegetation. Monthly herbaceous vegetation cover was calculated from herbaceous biomass outputs from the

Rangeland Production Model, and a simple linear regression relating standing biomass to vegetation cover estimated from vegetation collected in the study area as described by Ahlborn [38]. The temporal inputs were collected on a monthly basis, excepting wind speed, which is a daily data value that is used to calculate a total monthly wind factor. Due to the differences in resolution of each of our input data sources, each GeoTiff file was scaled to the resolution of the elevation file with Python, averaging values for groups of pixels for higher resolution data. Histograms of the five primary soil loss factors are available in the Supplementary Materials Section (Figures S2–S6).

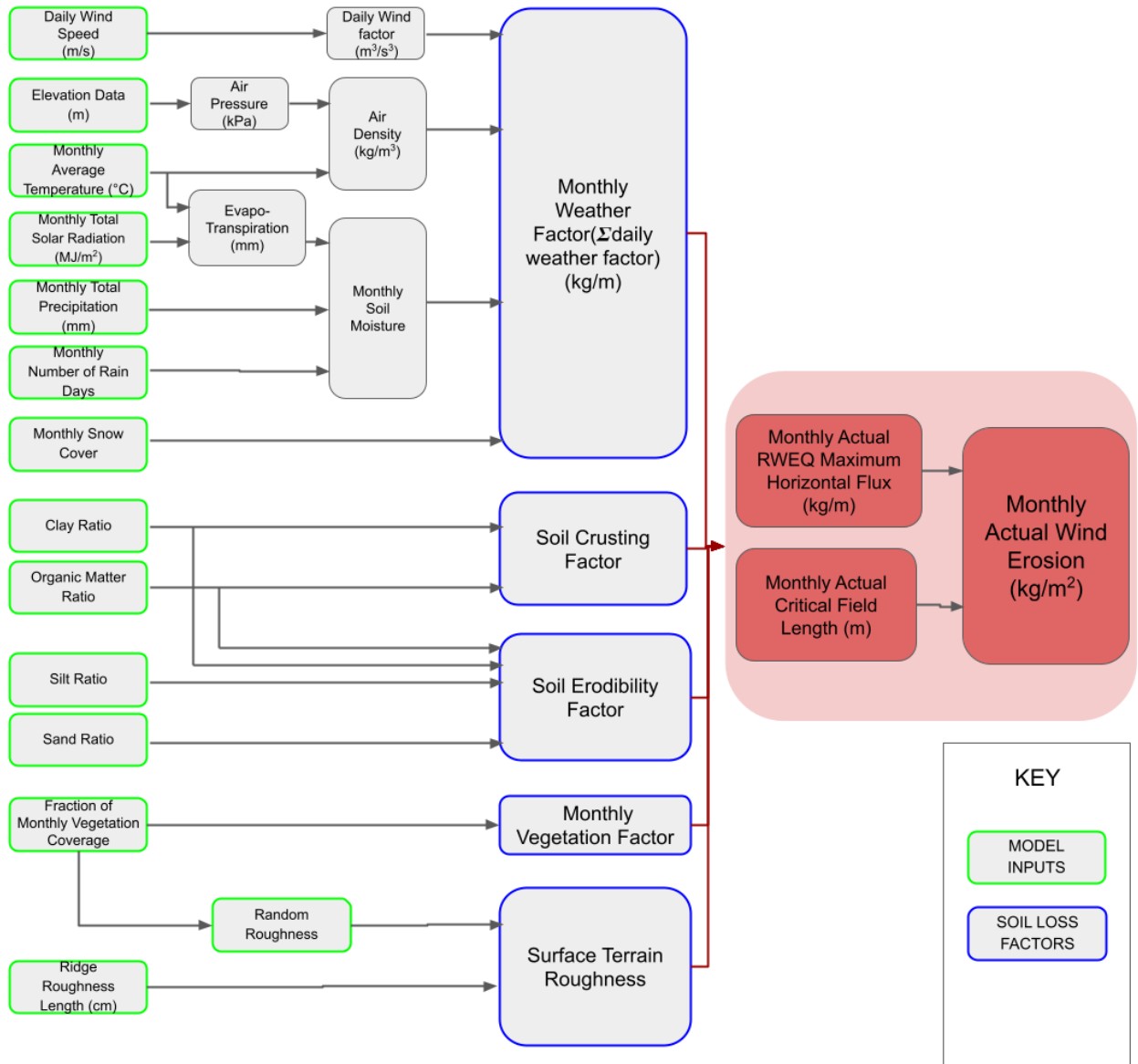

**Figure 2.** Model flowchart summary. Demonstrates how inputs are utilized to calculate factors used for final soil loss calculation.

Table 1 summarizes the data used for this paper, including the source, unit, original resolution, and format.

**Table 1.** Data collection.

| Category | Dataset | Source | Temporal Range (Year) | Unit | Original Resolution | Format |
|---|---|---|---|---|---|---|
| Elevation | Elevation Data (DEM) | SRTM | 2000 | m | 0.05985 degrees$^2$ per pixel | GeoTIF |
| Climate | Monthly Average Temperature | Worldclim current conditions | 1970–2000 | °C | 0.00833 degrees$^2$ per pixel | GeoTIF |
| | Monthly Total Precipitation | Worldclim current conditions | 1970–2000 | mm | 0.00833 degrees$^2$ per pixel | GeoTIF |
| | Monthly Solar Radiation | Rangeland Production Model Output | 2014–2015 | MJ/m$^2$ | 0.00324 degrees$^2$ per pixel | GeoTIF |
| | Monthly Snow Cover | Rangeland Production Model Output | 2014–2015 | % (probability) | 0.05003 degrees$^2$ per pixel | GeoTIF |
| | Monthly Number of Rain Days | Amicus Travel | 2020 | day | 0.05985 degrees$^2$ per pixel | GeoTIF |
| | Daily Wind Speed | Copernicus Climate Data Store | 1981–2019 | m/s | 0.00833 degrees$^2$ per pixel | GeoTIF |
| Soil | Sand Ratio | ISRIC Soil Grids | 2014–2016 | % | 0.00208 degrees$^2$ per pixel | GeoTIF |
| | Silt Ratio | ISRIC Soil Grids | 2014–2016 | % | 0.00208 degrees$^2$ per pixel | GeoTIF |
| | Clay Ratio | ISRIC Soil Grids | 2014–2016 | % | 0.00208 degrees$^2$ per pixel | GeoTIF |
| | Organic Matter Ratio | Rangeland Production Model Output | 2014–2016 | % | 0.00324 degrees$^2$ per pixel | GeoTIF |
| Vegetation | Monthly Fraction of Vegetation Coverage | Rangeland Production Model Output | 2014–2016 | % | 0.05003 degrees$^2$ per pixel | GeoTIF |

*2.3. Sensitivity Analysis*

In the smaller study area in the Gobi desert, we explored the sensitivity of soil loss to each input by varying their value, and recomputing soil loss. Specifically, each model input (wind speed, temperature, etc.), and each of the five intermediate factors (weather factor, erodible fraction, etc.) were multiplied by a scaling factor (50%, 75%, 90%, 100%, 110%, 125%, 150%). For each parameter, in addition to the actual soil loss, three additional estimates were done with three different theoretical vegetation coverage scenarios: uniform 0%, 50%, and 100% vegetation cover. For these theoretical vegetation coverage scenarios, fractional vegetation coverage rasters were pre-computed by assigning every pixel within the area of interest a value of 0, 50, and 100, respectively, as the equations express a percentage representation. The soil loss was summed temporally as well as spatially to generate a single value representation of the scaling effect for each parameter. Line graphs were generated for each model input to show the scaling effect.

In addition, a percentage change of the annual soil loss compared to the actual annual soil loss was computed for six different theoretical vegetation coverage scenarios. Three of the scenarios used absolute, uniform fractional vegetation coverage of 0%, 50%, and 100%. The other three scenarios used 110%, 120%, and 150% of the original values for each pixel. Soil loss maps were generated for each month and then summed temporally to generate annual soil loss maps for visual comparison.

Another sensitivity analysis was done by computing the percentage change of the annual soil loss compared to the actual annual soil loss at the maximum and minimum values for each model input. The computation was done by setting only one input to the

maximum or minimum at a time. For each analysis, each pixel in the area of interest for the selected input parameter was set uniformly to the maximum or minimum value. The soil loss was summed temporally as well as spatially to have a single value representation of the scaling effect for each parameter. Results are captured in Table 2 for comparison.

**Table 2.** Sensitivity analysis of soil loss (*SL*) against model inputs. Sensitivity was measured as the perfect change in the annual, spatial soil loss sum if an input increased or decreased by a percentage of the original value, or was uniformly set to the minimum or maximum value of the spatial data. The actual annual soil loss rate was 8800 t/km$^2$.

| Variable | Affected Soil Loss Factors | Global Average of Variable | Global Range {Min–Max} | Maximum Spatial Range {Min–Max} | Maximum Seasonal Range {Min–Max} | SL Sum (%) Difference with Variable = 50% of Actual Value | SL Sum (%) Difference with Variable = 150% of Actual Value | SL Sum (%) Diff When Variable = Min of Global Range Uniformly | SL Sum (%) Diff When Variable = Max of Global Range |
|---|---|---|---|---|---|---|---|---|---|
| Wind_speed (m/s) | *WF* | 1.2 | 0–13 | 0.7–17 | NA | −100 | 309 | −100 | 322 |
| Temp (°C) | *WF* | −0.13 | −24–20 | 5.8–7.4 | 33–40 | −3.06 | 2.70 | 52.5 | −9.43 |
| Precip (mm) | *WF* | 8.8 | 0–47 | 2–28 | 19–46 | 2.61 | −1.92 | 5.60 | −2.91 |
| Precip days | *WF* | 0.75 | 0–3 | NA | 0–3 | 2.61 | −1.92 | 5.60 | −17.7 |
| Solar_rad (MJ/m$^2$) | *WF* | 667 | 260–1050 | 0.58–42 | 748–789 | −2.79 | 1.69 | −3.17 | 0.73 |
| Sand (ratio) | *EF* | 0.43 | 0.32–0.50 | 0.19–0.19 | NA | −16.6 | 15.5 | −8.70 | 4.66 |
| Silt (ratio) | *EF* | 0.39 | 0.32–0.48 | 0.16–0.16 | NA | −7.52 | 7.30 | −2.68 | 3.39 |
| Clay (ratio) | *EF*, *SCF* | 0.18 | 0.14–0.25 | 0.11–0.11 | NA | 75.0 | −46.1 | 27.2 | −42.2 |
| SOM (%) | *EF*, *SCF* | 2.7 | 2.3–3.2 | 0.88–0.88 | NA | 11.6 | −13.4 | 3.56 | −4.65 |
| *fvc* (%) | *COG*, *K'* | 25 | 14–49 | 13–34 | 0–30 | 10.3 | −17.4 | 8.59 | −50.3 |
| *WF* (kg/m) | | 25 | 0–520 | 22–510 | 15–516 | −52.5 | 43.4 | NA [1] | NA [1] |
| *EF* | | 0.43 | 0.39–0.45 | 0.05–0.05 | 0.00–0.00 | | | | |
| *SCF* | | 0.18 | 0.10–0.25 | 0.15–0.15 | 0.00–0.00 | | | | |
| *K'* | | 0.79 | 0.48–0.86 | 0.07–0.37 | 0.00–0.35 | | | | |
| *COG* | | 0.89 | 0.79–0.94 | 0.06–0.14 | 0.00–0.12 | | | | |

Note [1]: The sensitivity analysis for the soil loss factors (*WF*, *SCF*, *EF*, KPRIME, *COG*) was done by applying a single scale factor to the product of these factors in the RWEQ model for maximum transported capacity by wind (*Qmax*) and the critical field length (*S*). As such, there was no representative min or max value that could be used.

## 3. Results and Discussion

### 3.1. Model Results in Eastern Gobi Steppe

The results from our model are consistent with previous studies done in the same area.

A RWEQ-based wind erosion analysis was done over the Inner Mongolia province of China [16]. The equations were identical, excepting the equation for potential evapotranspiration. Our coefficient, which multiples solar radiation and temperature after combining the numerical factors in the equation, was 0.00531, while the Zhang coefficient was 0.000277 [16]. Our equation to calculate potential evapotranspiration (*ETp*) was derived from Samani and Pessarakli [27]. After comparison, it was found that the *ETp* values were consistent and the differences in equation would not affect the correctness of final results.

Our area of interest, the Eastern Mongolian Steppe, overlaps part of the region covered by their study. That paper categorized wind erosion into different classes of severity from tolerable to destructive. A visual comparison of their figures showed that the overlapping area falls into the severe and very severe categories. According to Zhang [16], the severe category has a soil erosion range of 5000 to 8000 t/km$^2$/a and the very severe category has a soil erosion range of 8000 to 15,000 t/km$^2$/a. Our estimated total soil erosion was $61 \times 10^{10}$ kg over an area of $69.3 \times 10^9$ m$^2$, which corresponds to 8800 t/km$^2$. To give a comparison, that amount of soil loss for an area of the size of the city San Francisco would be three elephants in weight. Considering that the bulk density of fine soil that is transported in wind erosion is 1500 kg/m$^3$, the average thickness of soil transported in a square kilometer is about 0.5 cm [39]. Realistically, not all the total area will be eroded by the wind, because the transported soil can accumulate in certain locations, e.g., in the surroundings of a windbreak, a vegetation patch, or in a ditch. This shows that the results of this study are consistent with prior published results in the overlapping areas of Inner Mongolia.

Table 3 summarizes the categories of severity of wind erosion from Zhang [16] and compares them with our annual wind erosions based on each quadrant of the area of interest (Figure 3).

**Table 3.** Data comparison.

| Area | Erosion Level as Defined by Zhang [16] (t/km$^2$) | Annual Wind Erosion, as Calculated in This Study (t/km$^2$) |
|---|---|---|
| Total Soil Loss/Total Area | Very Severe (8000–15,000) | 8800 |
| Quadrant I (North East) | Moderate (2500–5000)–Destructive (>15,000) | 2600–63,000 |
| Quadrant II (North West) | Very Severe (8000–15,000)–Destructive (>15,000) | 8900–69,000 |
| Quadrant III (South West) | Very Severe (8000–15,000)–Destructive (>15,000) | 8200–93,000 |
| Quadrant IV (South East) | Destructive (>15,000)–Destructive (>15,000) | 21,000–95,000 |

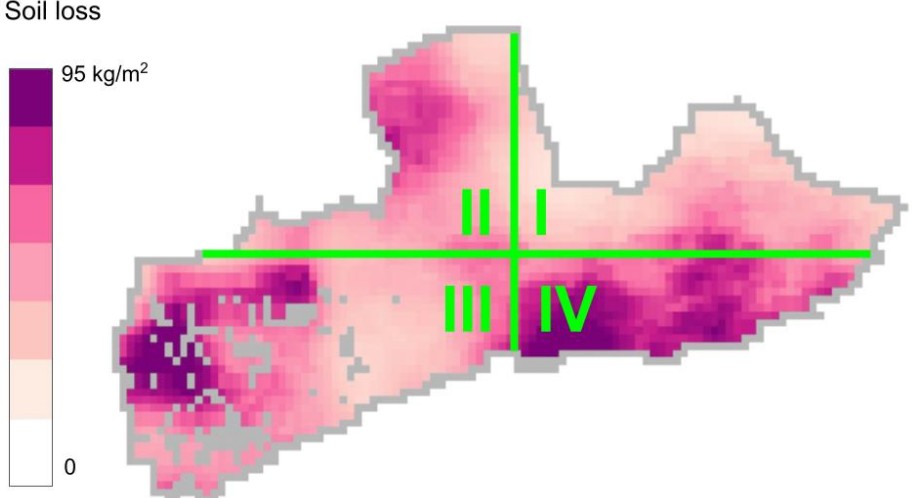

**Figure 3.** Map of annual soil erosion in our area of interest split into four quadrants corresponding with rows of Table 3. For each quadrant, the spatial maximum and minimum soil erosion are represented as categories according to Zhang [16].

The Eastern Gobi Steppe Area has almost double the soil loss per unit area of Inner Mongolia because it is less vegetated.

Our model estimates the total annual soil loss for the Mongolian Eastern Gobi Steppe to be $0.61 \times 10^9$ t. Another study on soil loss in Inner Mongolia of China, a nearby region, estimated the annual soil losses to be in the range of 4 to 5.7 billion tonnes [15] In comparison, our model estimates the total annual soil loss for the Mongolian Eastern Gobi Steppe to be $0.61 \times 10^9$ t ($61 \times 10^{10}$ kg). The area of interest in that study was 1.2 million km$^2$, whereas our area of interest had an area of 69,300 km$^2$. Comparing these numbers, we see that the Eastern Gobi Steppe Area has almost double the soil loss per unit area of Inner Mongolia. This is because the Inner Mongolia area of interest is larger and contains areas with a large range of vegetation coverage, so there is less wind erosion on average due to the influence of vegetation-heavy areas, which are not present in the Eastern Gobi Steppe.

### 3.2. Environmental Factors Contributing to Soil Loss

### 3.2.1. Impact of Five Primary RWEQ Factors

Soil erosion is affected by the seasonal variation of the weather factor and vegetation coverage.

The RWEQ's primary factors are weather factor (*WF*), vegetation factor (*COG*), surface terrain roughness (*K′*), erodible fraction (*EF*), and soil crusting factor (*SCF*). Figure 4 displays maps of each of these five primary factors, along with the estimated soil loss, for our area of interest. Though the spatial distribution of each factor varies across months, we have chosen four months, January, April, July, and October, to represent the four seasons. In addition to the primary factors, fractional vegetation coverage maps are shown to highlight the relation between seasonal soil loss and vegetation coverage. The weather factor represents the effect of climate on wind erosion, and combines wind speed, soil moisture, and snow cover. As shown in Figure 4A,E, the higher weather factor in January leads to a higher soil loss for the month. Figure 4C also shows that *COG* is high for January. *COG* is inverse to fractional vegetation coverage, which implies lower vegetation coverage. Indeed, the fractional vegetation coverage in January is the lowest of the four seasons, which contributes to the high soil loss in that month. Increased wind speeds caused increased soil erosion. Whereas increased soil moisture, vegetation, and snow cover decrease wind sensitivity. Thus, these factors must be lower for wind to cause severe soil loss. The lower levels of vegetation coverage corroborate this conclusion. In October, the low weather factor and high fractional vegetation coverage boost each other's effect in decreasing wind sensitivity, resulting in comparatively lower soil loss in that month. *K′* shows similar variability across the seasons to *COG* since both factors are dependent on fractional vegetation coverage. Any impact of cultivation practices on *K′* was not taken into account in this study due to lack of data. The seasonal variation of *SCF* and *EF* is not expected in this region and this project used non-temporally variable soil composition data.

### 3.2.2. Sensitivity of Soil Loss

Sensitivity of the soil loss to the environmental inputs was measured via analysis of percentage change of soil loss caused by percentage variation of each environmental input.

Soil loss was most sensitive to changes in wind speed, as expected when using the RWEQ, which emphasizes the effect of wind.

Of all inputs to the RWEQ model, sensitivity of soil loss to wind speed was greatest (Table 2). A couple of results deserve special attention. When the wind speed for every day was set to the maximum wind speed spatially, the soil loss increased by 322% (first row, Table 2). In our area of interest, the wind speed values were mostly less than the threshold, hence when the value was forced to the maximum, there was a significant increase in soil loss. This observation was more pronounced when all spatiotemporal wind speed values were increased by 150%. As most of the wind speed values were between 3 m/s and 5 m/s, a 150% increase made many of them higher than the threshold and contributed to the soil loss. In addition, the wind speed has a cubic relation to the weather factor, so a small

increase in non-zero entries of wind speed values resulted in a significantly higher weather factor, hence, soil loss. This is in line with the general principle of the RWEQ's reliance on wind as a cornerstone [12].

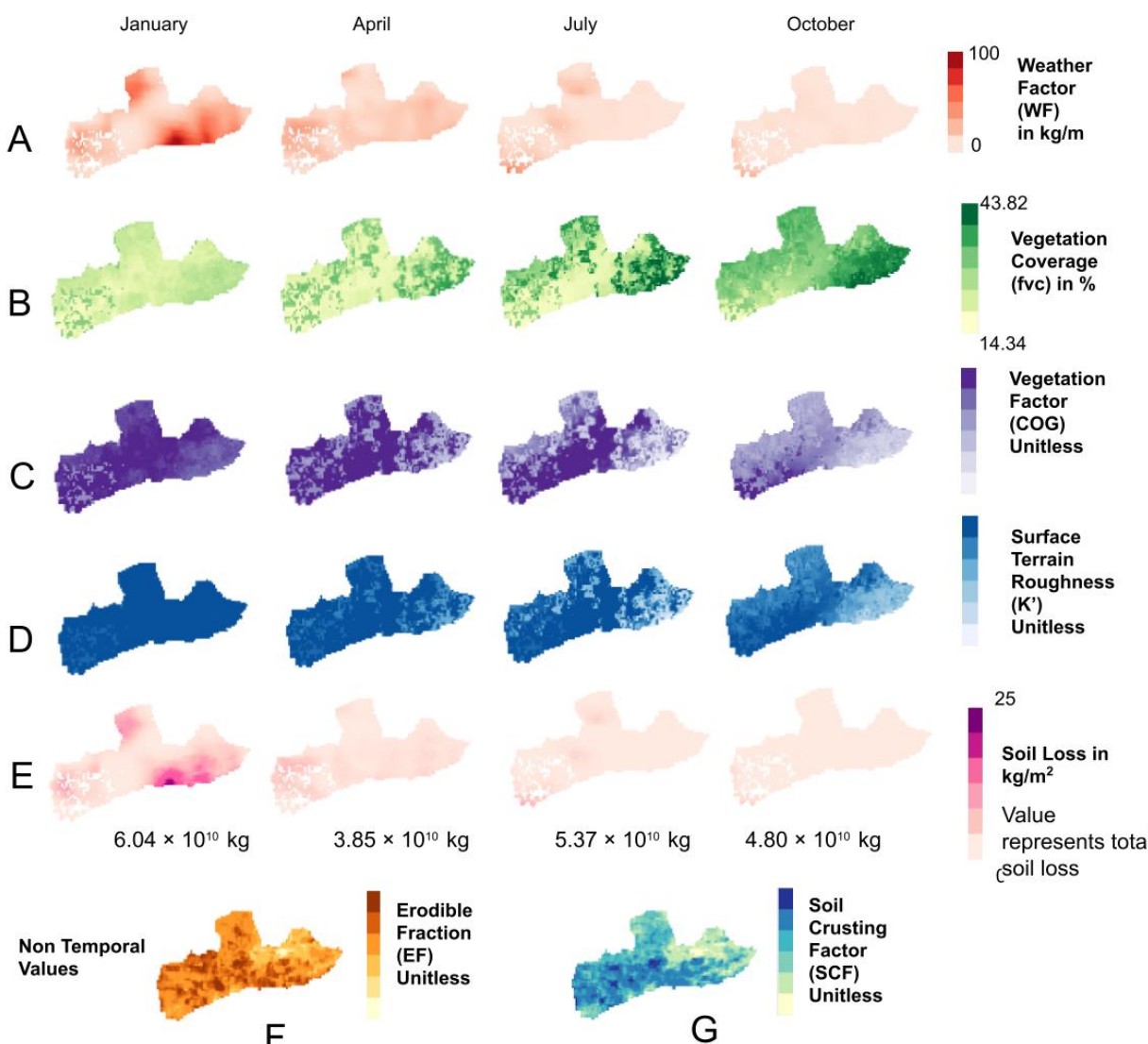

**Figure 4.** Soil loss factors per season. Inputs are monthly; this figure highlights one month per season. Each row and map labelled with a letter represents a primary soil factor, soil loss, or vegetation coverage. Each column represents a season by displaying the map for one month.

Fractional vegetation and clay ratio are the most significant soil-based factors.

The soil parameters (sand, silt, clay, organic matter) also have significant effects on soil loss (Table 2); in particular, the clay ratio especially affects soil loss as higher clay in the soil allows it to maintain larger clumps which could reduce the impact of wind erosion [40]. Similarly, fractional vegetation coverage also has a significant effect on soil loss.

Precipitation and solar radiation are not as significant.

The results show a lack of sensitivity of soil loss to precipitation, number of rain days, and solar radiation, as shown by the less than 10% change in the corresponding rows of Table 2. With a range of only 0 to 47 mm of rain both spatially and temporally, the low sensitivity may be a result of low precipitation (Table 2). The difference in sensitivity between these factors is displayed even more clearly in the plots of Figure 5, where there are steep changes in the plot for factors such as wind speed, and flatter lines for precipitation and solar radiation.

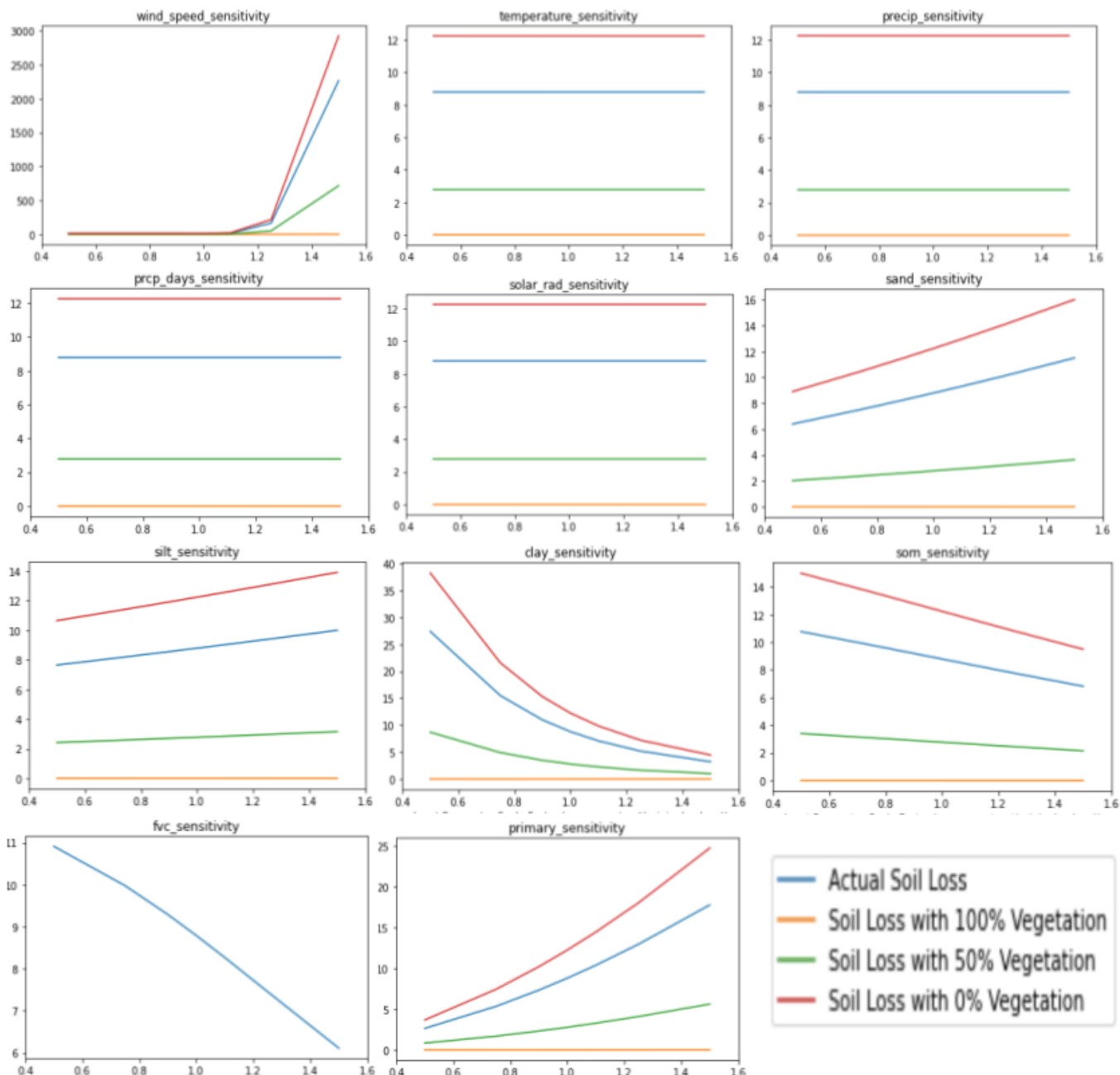

**Figure 5.** Plots of sensitivity of soil loss to model inputs. The sensitivity of soil loss was plotted against scenarios of 0%, 50%, 100%, and original vegetation coverage. The *y*-axis represents soil loss in kg/m$^2$ and the *x*-axis represents the scale parameter that multiplies the input (e.g., 0.5 for 50%).

### 3.3. Impact of Vegetation Coverage

3.3.1. Effect of Vegetation Coverage on Soil Loss

Increasing vegetation coverage significantly reduces wind erosion.

Vegetation coverage has a significant impact on soil loss, as observed by the soil loss percentage change with varying magnitudes of fractional vegetation coverage (Figure 5). The areas with minimal vegetation coverage (Map B) showed more soil loss (Map A), such as the western side of the area of interest. At fractional vegetation coverage in the range of 30% to 50%, the soil loss was in the range of 0.001 kg/m$^2$ to 0.002 kg/m$^2$, as shown in the eastern side of the area in Maps A and B. A potential soil loss estimation with 0% fractional vegetation coverage (*FVC*) shows more soil loss, as shown in Map C. Specifically, the eastern side showed a higher percentage change in soil loss in the range of 70% to 100%

(Map C). This is consistent with the higher *FVC* in that area (Map B), where the soil loss was minimized by the higher vegetation coverage. If that coverage were removed, the area becomes susceptible to higher soil loss. On the other hand, the theoretical scenarios where *FVC* was uniformly 50% (Map D) and 100% (Map E) indicate that soil loss could be significantly reduced with substantially increased vegetation. Soil loss would decrease by 60% to 80% with 50% *FVC* and decrease by 90% to 100% with 100% *FVC*. If the vegetation coverage in each area were increased to 110% (Map F), 120% (Map G), or 150% (Map H) of the original vegetation coverage, it would result in reduced soil loss to a lesser degree, from 20% to 60% decrease.

### 3.3.2. Vegetation Coverage Management

Increasing vegetation coverage seems like an effective way of decreasing wind erosion.

Based on our six vegetation cover scenarios (Figure 6), we show that increasing the original vegetation cover by only 10% or 20% is needed to achieve a 13% to 25% decrease in soil loss. The current average annual vegetation coverage is 25%, which means that the vegetation coverage must be doubled (increased by a factor of 1.4 to 3.3 depending on area) to enable the more drastic reductions in soil loss (60% to 80%) that would be achieved with a uniform vegetation cover of 50%. As reported by the sensitivity analysis of Table 2, a lack of sensitivity was observed for climate factors such as precipitation, solar radiation, and temperature.

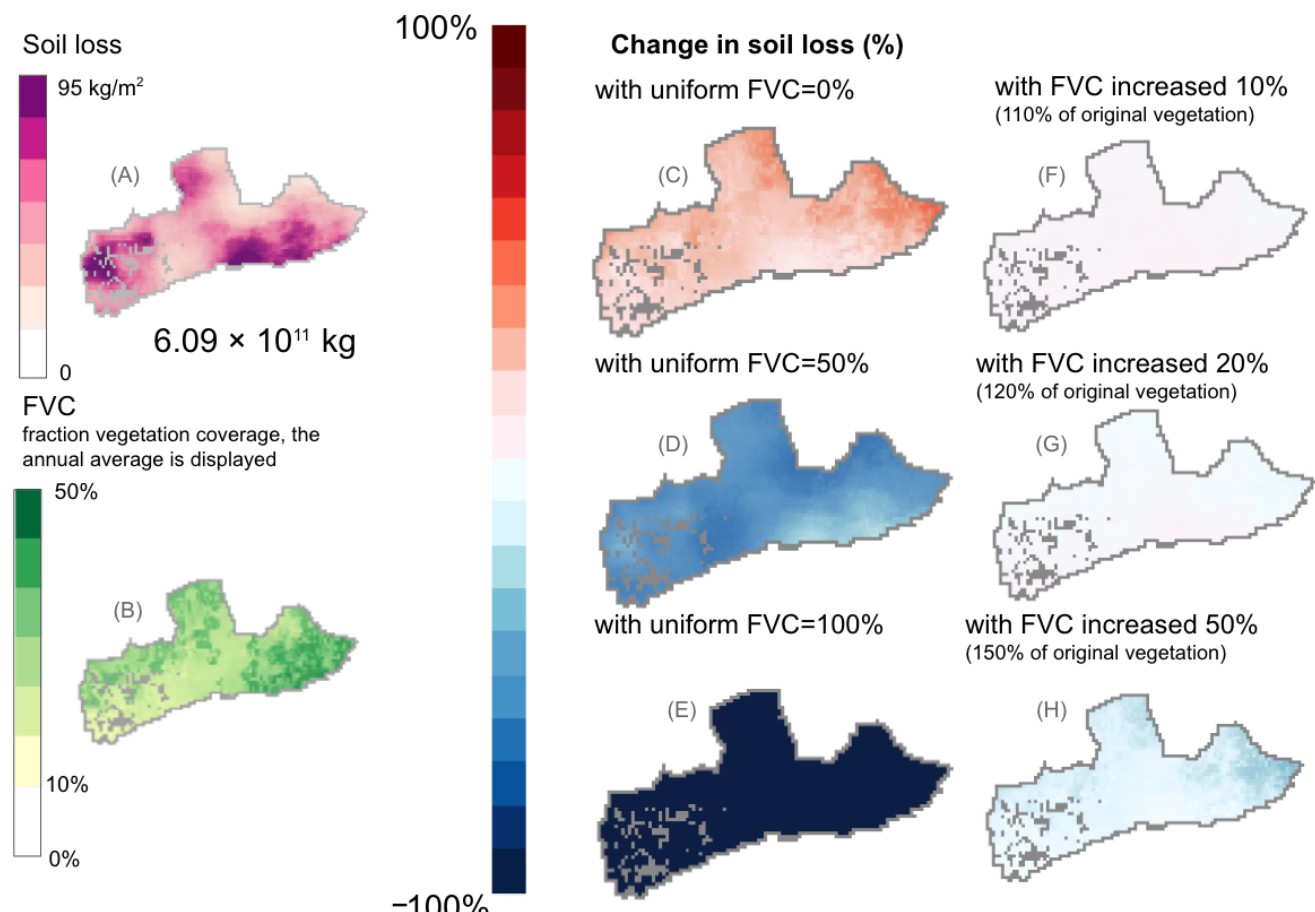

**Figure 6.** Soil loss with current vegetation, and six scenarios of vegetation cover. Figure (**A**) represents annual soil loss. Figure (**B**) represents the annual average fractional vegetation coverage. Figures (**C**) through (**H**) represent percent change in annual wind erosion with different hypothetical fractional vegetation coverages.

Protection and increase of vegetation coverage is vital to decreasing wind erosion.

These results highlight the importance of safeguarding vegetation coverage as it can have a significant contribution to ecosystem service preservation by preventing wind erosion. A similar conclusion was reached by Jiang et. al. [15]: promoting forest and grassland restoration from farmland, restricting grassland cultivation, limiting the number of livestock heads on grassland, and allowing a timely recovery for degraded grassland can lead to reduced soil loss. A possible follow-up study of our research could explore the type of vegetation that causes the greatest reduction in soil loss, similar to the study done by Zhang et al. on the impact of different groundcover plants in combating soil erosion [41,42].

These results suggest that while future climate change may bring higher wind speeds to the region, increasing vegetation cover is a powerfully effective tool to reduce soil loss. This finding suggests that protecting and increasing vegetation coverage would be effective in decreasing wind erosion.

## 4. Conclusions

Our results show that the RWEQ-based model that we developed is applicable to estimating soil loss in the Mongolian Eastern Gobi Steppe, as our estimated total soil erosion was consistent with prior published results in neighboring areas. Our findings indicate that in addition to the weather and soil-related model inputs, the vegetation coverage has a significant impact on soil loss. Indeed, our sensitivity analysis shows that varying vegetation coverage results in the second largest percentage change in soil loss, following wind speed. This is especially significant for this area as wind speeds are often below the threshold for wind erosion. The estimated soil loss was lower in the eastern part of the area of interest, which had a higher vegetation coverage. The spatial soil loss estimates using different theoretical vegetation coverages show that increasing fractional vegetation coverage can reduce soil loss. Even an increase of 10% to 20% can have measurable reductions in soil loss. As demonstrated by Table 2, changes in wind speed and soil composition also have significant impact on soil loss.

Government efforts can also be extremely beneficial to lessening wind erosion. If a project such as the aforementioned Beijing−Tianjin Sandstorm Source Control Project is implemented in our area of interest, the Eastern Gobi Steppe, increasing vegetation coverage to decrease wind erosion, valuable ecosystem services can be preserved.

Directions for future research.

It is important to further investigate the impact of climate change on future wind erosion. In the near term, higher temperatures and more variable precipitation are expected to drive large changes in vegetation cover in this region [37]. Our model is well suited to show the impacts of these changes on future wind erosion. As a preliminary demonstration, we used our RWEQ based model to explore the percentage change in soil loss under future conditions of vegetation cover (see Supplementary Materials). Our analysis did not include future wind speed data, however, which limited the usefulness of these preliminary results. Scenario analysis is a promising avenue of future research: further analysis of future soil erosion should be done with predicted wind speed data for this region.

**Supplementary Materials:** The following supporting information can be downloaded at: https://www.mdpi.com/article/10.3390/land11081204/s1, Figure S1. Percentage change in soil loss from current data to a future extreme climate scenario; Figure S2. Monthly weather factor (*WF*); Figure S3. Erodible fraction (*EF*); Figure S4. Soil crusting factor (*SCF*); Figure S5. Monthly surface terrain roughness (*K'*); Figure S6. Monthly vegetation factor (*COG*); Table S1. Correlation of model inputs to weather factor.

**Author Contributions:** Conceptualization, C.W.; data curation, C.W., B.H. and V.A.K.; investigation, I.N.T.; methodology, I.N.T.; project administration, C.W.; software, I.N.T. and B.H.; supervision, C.W.; validation, I.N.T.; writing—original draft, I.N.T.; writing—review and editing, C.W., B.H. and V.A.K. All authors have read and agreed to the published version of the manuscript.

**Funding:** This work was supported by the NASA Biodiversity and Ecological Forecasting Program, award Number NNX17AG56G.

**Institutional Review Board Statement:** Not applicable.

**Informed Consent Statement:** Not applicable.

**Data Availability Statement:** The input data used is available as sample data in this repository: https://github.com/charlottegiseleweil/winderosion (accessed on 30 June 2022). Following the usage example in the README will generate the wind erosion outputs used in this study.

**Acknowledgments:** The authors would like to thank Rebecca Chaplin-Kramer, who inspired this work, made it possible and gave great advice all along.

**Conflicts of Interest:** The authors declare no conflict of interest. The funders had no role in the design of the study; in the collection, analyses, or interpretation of data; in the writing of the manuscript, or in the decision to publish the results.

**Software Availability:** The Python code of the model implemented in this paper is available in: https://github.com/charlottegiseleweil/winderosion (accessed on 30 June 2022). The README file includes a usage example, as well as a link to a sample data set.

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
