# Peer review of "Vegetation Drastically Reduces Wind Erosion: An Implementation of the RWEQ in the Mongolian Gobi Steppe"

_land, doi:10.3390/land11081204_

Round 1

Reviewer 1 Report

That is a very interesting work of You which may be of interest of the scientific public. It is also very important from the view of global change. 

What the Authors should improve the citations of the study, the presentation of the study site (because that misess a lot of concrete data about climatic and soil conditions of the area). Also theres are some confusing remarks in Discussion subchapter.

In the Coclusion subchapter, there are some pharagraps which suit better into the Introduction and also Discussion subchapter.

My remarks and questions are available into the attached document.

Reviewer 2 Report

Comments-questions (to be clarified in the text):

1. Line 175: What exactly is the "chain random roughness"?

2. The titles of the sections, e.g. 3.1, 3.2, 3.2.2, 3.3, 3.3.2, should not be sentences (in my opinion).

3. All figures and tables should be cited in the text, e.g. Figure 3.

4. The references should be written in a uniform way.

5. The written presentation of the article should be improved considerably. See annotated manuscript! 

6. Some of the supplementary material could be included in the main text. 

Round 2

Reviewer 2 Report

1. Sub-section 2.1: RWEQ Model: The impact of the factors included in RWEQ model on wind erosion should be explained from physical point of view.

2. The written presentation of this study should be improved considerably.

3. See annotated manuscript!
